# The ESCRT Machinery: Remodeling, Repairing, and Sealing Membranes

**DOI:** 10.3390/membranes12060633

**Published:** 2022-06-19

**Authors:** Yolanda Olmos

**Affiliations:** Department of Cell Biology, School of Biology, Complutense University of Madrid, 28040 Madrid, Spain; yolmos@ucm.es; Tel.: +34-913945128

**Keywords:** ESCRT, membrane scission, reverse topology, nuclear envelope, lysosome, membrane sealing, membrane repair

## Abstract

The ESCRT machinery is an evolutionarily conserved membrane remodeling complex that is used by the cell to perform reverse membrane scission in essential processes like protein degradation, cell division, and release of enveloped retroviruses. ESCRT-III, together with the AAA ATPase VPS4, harbors the main remodeling and scission function of the ESCRT machinery, whereas early-acting ESCRTs mainly contribute to protein sorting and ESCRT-III recruitment through association with upstream targeting factors. Here, we review recent advances in our understanding of the molecular mechanisms that underlie membrane constriction and scission by ESCRT-III and describe the involvement of this machinery in the sealing and repairing of damaged cellular membranes, a key function to preserve cellular viability and organellar function.

## 1. Introduction

Eukaryotic cellular membranes are highly dynamic entities that undergo continuous remodeling, fusion, budding, and fission events that are essential to cell and tissue viability. The endosomal sorting complex required for transport (ESCRT) machinery has been identified as a key player in an increasing number of these membrane-remodeling events. ESCRTs have the unique ability to catalyze membrane fission from within membrane necks, in opposition to the well-characterized formation of coated vesicles, where fission occurs from the vesicle neck exterior [1]. This ‘reverse’-topology membrane scission constitutes the primary biochemical function of ESCRTs and allows the constriction and scission of membrane necks and the repair of membrane fenestrations; for example, when the plasma membrane is punctured or damaged. Over the past recent years, a myriad of cellular functions for the ESCRT machinery have been described. Functions range from the formation of multivesicular bodies (MVBs) in the endosomal sorting pathway [2], virus budding [3], and cytokinetic abscission [4], to nuclear envelope surveillance and reformation [5,6,7], autophagosome closure [8], and plasma-membrane [9] and lysosome membrane repair [10,11]. ESCRT biology and functions are described in detail in comprehensive recent reviews [12,13,14,15]. In the present review we will focus on the molecular mechanisms of this machinery, highlighting ESCRT roles in membrane remodeling, repair, and sealing.

## 2. Membrane Remodeling by ESCRTs

Found in *Archaea* [16,17,18], ESCRTs are highly conserved through evolution. Here we will mainly focus on mammalian cells and yeast, where the ESCRT machinery comprises four multimeric protein core complexes termed ESCRT-0, ESCRT-I, ESCRT-II, and ESCRT-III, plus the AAA ATPase VPS4 and additional accessory proteins (Table 1). Bacteria and *Archaea* express ESCRT-III proteins but lack ESCRT-0, -I, and -II components [19,20]; HRS and STAM (ESCRT-0) are not found in plants, but ESCRT-I to -III are conserved [21,22,23].

Most ESCRT-mediated functions require a topologically equivalent reverse membrane remodeling for their completion. In addition, ESCRTs can carry out normal topology scission, from the outside of membrane necks [24,25,26]. ESCRTs constitute, therefore, a highly versatile remodeling machinery, and their mechanism of action has attracted a great deal of research efforts over the past years.

Normally localized in the cytoplasm, ESCRT subunits are sequentially recruited by site-specific adaptor proteins to different membranes. For instance, ESCRT-0 is essential for consecutive recruitment of other ESCRT components to the endosomal membrane in multivesicular body biogenesis [27,28]; CEP55, SEPT9, and additional pathways recruit ESCRT-I to intercellular bridges to facilitate cytokinetic abscission [4,29,30,31]; and viral Gag proteins recruit ESCRT-I in retroviral egression from the plasma membrane [32,33]. Recruited early-acting ESCRT factors initiate membrane bending and nucleate the assembly of the downstream ESCRT-III components. ESCRT-III forms a membrane-interacting oligomeric filament that is thought to mediate the membrane remodeling event, eventually resulting in scission [1,34]. Not all ESCRT-mediated biological processes require all complexes, but ESCRT-III and VPS4 appear to be universally required.

### 2.1. ESCRT-III Structure and Assembly

There are eight ESCRT-III proteins in yeast, and twelve in humans, named charged multivesicular body proteins (CHMPs) (Table 1). CHMP4/Snf7, CHMP3/Vps24, and CHMP2/Vps2, together with VPS4/Vps4, were shown to be indispensable components of the filaments that mediate membrane remodeling [35,36,37,38]. CHMP7 performs specialized functions in nuclear envelope reformation and repair [39], whereas CHMP5 remains poorly characterized. Structural work revealed that all CHMP proteins share a core structure that is thought to adopt two different conformations, known as *open* or *closed* [40]. In their closed state, they form a four-helix bundle, with α1 and α2 helices forming a long hairpin, the shorter helices α3 and α4 packed against the hairpin, and helix 5 folding back and packing against the closed end of the helical hairpin, as shown by the crystal structures of CHMP3 [41,42] and IST1 [43,44]. In their open state, helices α2 and α3 merge, disrupting the interaction between helix α4 and the hairpin, as shown for CHMP1B [43,44] and truncated forms of CHMP4 [45,46]. An intermediate, semi-open conformation, has also been described for yeast Vps24 [47]. Importantly, ESCRT-III proteins in all three conformations seem to be able to assemble into filaments [40,43,44,47]. However, these filaments might show different abilities in membrane binding and flexibility [40]. Whereas the closed conformation does not display membrane binding interfaces [41,48,49] and is thought to result in more rigid filaments, the more extended open conformation displays extended membrane-binding interfaces, appears to be in the polymerization-competent state [12,41,50], and forms highly flexible filaments [46,51], which would potentially allow the binding to membranes of a wide range of curvatures.

In general, ESCRT-III polymers are curved and flexible, and most possess a membrane-binding interface [40]. They often form copolymers with other ESCRT-III subunits and can take a variety of shapes on membranes in vitro and in vivo, including rings, spirals, helices, and cones [14,44,52,53]. These morphologies have been well characterized in recent years through structural biology approaches, cryo-electron microscopy, and atomic-force microscopy [43,45,46]. Interestingly, recent data have shown that the ESCRT-I complex can also form helical filaments [54], suggesting that early -acting ESCRT factors might not merely be bridging adaptors, but can also be involved in membrane deformation and ESCRT polymerization.

### 2.2. The Role of VPS4

ESCRT-III-mediated processes crucially rely on the activity of the AAA ATPase VPS4, the only known ATP-consuming factor in the membrane-scission reaction mediated by ESCRT [36]. VPS4 is recruited to membranes in order to translocate and unfold ESCRT-III components [55]; this process is mediated through the binding of microtubule interacting and trafficking (MIT) domains to MIT-interacting motifs (MIMs) in ESCRT-III proteins [56,57]. VPS4 function is essential to recycle ESCRT-III filaments and ensure high cytosolic levels of ESCRT-III monomers. Importantly, it also allows the remodeling of the ESCRT-III filament during pre-scission stages ([58,59] and Section 2.3 below) and increasing evidence suggests that VPS4 can additionally play a more active and mechanical role in membrane constriction and scission [14,34,60].

### 2.3. Mechanism of ESCRT-III Membrane Remodeling

In the last few years, a great deal of research work has been performed in order to understand how membrane constriction and scission is mediated by the ESCRT-III machinery (reviewed in [40,61,62]). Important advances in the field were achieved by using in vitro reconstitution studies using purified ESCRT subunits [44,47,59,63,64,65]. These allowed investigation of the structures, molecular properties, and interplay of various ESCRT-III filaments. Models for ESCRT-III-mediated membrane scission have been divided into three main categories [1,62]: in the classic ‘dome’ models ESCRT-III polymerizes forming a spiraling membrane-bound filament with consecutively narrower rings, and opposing membranes are brought together by fusion on top of a constricted cone or dome, with the narrow end of the cone either pointing towards the vesicle or towards the cytoplasm [66,67]; in the ‘buckling/unbuckling’ models mechanical forces provided by tension-driven transitions between planar and helical ESCRT-III filament configurations allow tubule extrusion [52] or vesicle release [34]; finally, in the ‘protomer conversion’ models, filament constriction occurs in response to Vps4-mediated subunit turnover [53] or incorporation into the filament of additional ESCRT-III subunits with different properties [44]; this can change filament’s curvature and rigidity, leading to a rapid structural change and subsequent membrane neck constriction. The proposed models are not mutually exclusive and, in fact, recent studies have culminated in a unifying model that combines all these mechanisms to explain constriction and scission of membrane necks by ESCRTs [59].

Pfitzner et al. reconstituted ESCRT-III-Vps4 assembly on supported bilayers, liposomes, and within membrane tubules, and analyzed ESCRT-III subunit binding and release, membrane deformation, changes in ESCRT-III filament orientation, and ESCRT-induced membrane fission [59]. As a result, they proposed a mechanism of stepwise changes in ESCRT-III filament structure and mechanical properties via exchange of the filament subunits to catalyze ESCRT-III activity (Figure 1). In this model, the upstream ESCRT machinery nucleates Snf7, which polymerizes forming a single-stranded filament. This filament is thought to form first because it binds well to flat membranes and can be nucleated by early-acting ESCRT complexes in vivo [54,59,63]. The Snf7 filament then recruits a second filament containing the Vps2-Vps24 pair, which together recruit a third filament comprising Vps2 and Did2; Vps2-Did2 in turn recruits and is finally replaced by the Did2-Ist1 pair. The different biophysical properties of each ESCRT-III subunit results in heteropolymers that differ in their assembly, disassembly, recruiting, and membrane deformation properties. Vps2-Vps4 filaments have higher affinity for Snf7 filaments, whereas Vps2-Did2 filaments bind best when Snf7 and Vps2–Vps24 filaments are already present, explaining the recruitment order. Conversely, Vps2–Did2 filaments recruit Vps4 depolymerization activity better, which favors ESCRT-III disassembly. As mentioned above, Vps4 binds most ESCRT-III subunits and mediates their extraction and exchange, which is necessary for successful narrowing of the neck [36,60]. Moreover, Vps4 disassembles ESCRT-III filaments with different efficiencies, in the order Vps2-Vps24 > Snf7 > Vps2-Did2 > Did2-Ist1. This results in a unidirectional reaction pathway (Figure 1). Interestingly, the different filaments also show distinct membrane deformation activities. The exchange of Vps24 for Did2 bends the polymer-membrane interface, triggering the transition from flat spiral polymers to helical filaments and driving the formation of membrane protrusions. This ends with the formation of a tight Did2-Ist1 helix that constricts the tubule and is shown to be able to promote fission when bound on the inside of membrane necks. Vps4 activity is required not only for constriction but also to complete scission, probably playing a role in fission beyond the establishment of the Did2/Ist1 polymer.

With this model Pfitzner et al. established the common principles of a general mechanism by which ESCRT-III remodels membranes: a sequence of subunit exchanges that switches the architecture and mechanical properties of ESCRT-III filaments. The ESCRT field seems now to be converging on this consensus mechanistic model. Additional recent work has provided for the first time direct evidence of spontaneous Snf7 spiral buckling using HS-AFM approaches [68]. However, further investigations will be necessary to answer some of the key questions that still remain open. For instance, cryo-EM of scission-capable complexes in the reverse topology process is needed to fully understand the ESCRT mechanism. It is also essential to clarify how the same complex (Did-Ist1) can induce fission in different orientations, assembling around or inside membrane necks [24,43]. Studies using high spatial resolution to address the directionality of filament growth will also be of interest. It is also worth considering that, like in dynamin-mediated membrane fission [69], additional external forces, including cargo crowding, might also be required to finalize the progression from highly constricted membrane structures towards fission.

## 3. ESCRTs in Membrane Sealing

In addition to its classic role in membrane remodeling and budding vesicles away from the cytoplasm, ESCRTs are also now understood to carry out additional important functions in membrane sealing and integrity maintenance.

The plasma membrane, which separates the cell from its surroundings, and the endomembranes enclosing the various cellular organelles, ensure the compartmentalization of eukaryotic cells. This is essential for their viability and functions. It is thus crucial for these membranes to remain intact so only gated transport of molecules and ions can occur through them. Therefore, mechanisms that mediate sealing of membrane holes are necessary, both during organelle biogenesis and as a response to membrane damage. The ESCRT machinery has been shown to play an essential role in sealing small membrane holes during the biogenesis of two organelles, the nucleus and the autophagosome [70,71].

### 3.1. Sealing of the Reforming Nuclear Envelope

The nuclear envelope (NE) is a double-layered membrane that encloses the nuclear genome and transcriptional machinery. In eukaryotic dividing cells, the NE completely disassembles during mitosis, so the nuclear compartment needs to be re-established at the end of each cell division [72]. During late anaphase, a new NE starts to reassemble around each of the two separated chromosome clusters to form daughter nuclei. This reassembly requires recruiting membranes from the endoplasmic reticulum, reconstituting nuclear pores, and severing microtubule connections between chromosomes and the spindle organizing centers [73]. At telophase, and in coordination with the removal of spindle microtubules, the reformed NE must seal remaining small holes to reestablish proper separation of the genome from the cytoplasm. Over the last few years, key studies have implicated the ESCRT machinery in this process [6,7,39,74,75].

ESCRT-III and VPS4 were shown to be transiently recruited to gaps in the reforming NE, where assembly of core subunits occurs in a canonical fashion, with CHMP4, CHMP3, and CHMP2 proteins recruited sequentially [76]. In the absence of successful ESCRT assembly, postmitotic nuclear envelopes have unsealed holes and are functionally ‘leaky’, leading to DNA damage at the nuclear periphery [6,7]. ESCRT-III thus plays an essential role in both generating and maintaining nucleocytoplasmic compartmentalization and protecting the genome from cytoplasmic insults.

The specific adaptor that recruits ESCRT-III to the reforming nuclear envelope is CHMP7 (Chm7 in yeast) [11], a hybrid ESCRT-II/ESCRT-III-like protein with an ER-localizing and membrane-binding motif in its N-terminal domain [39]. CHMP7 is engaged by the inner nuclear membrane protein LEM2 (ortholog of yeast Heh1/Heh2) and is essential for recruiting downstream ESCRT-III components to this organelle to effect nuclear membrane sealing [39,75,77] (Figure 2). Additionally, the ESCRT-III subunit IST1 is able to recruit the microtubule-severing enzyme Spastin to depolymerize microtubules and coordinate spindle disassembly with sealing of the NE [7] (Figure 2).

LEM2 bridges the NE with the underlying chromatin through an N-terminal LAP-2-emerin-MAN1 (LEM)-domain. LEM2 also contains a C-terminal winged helix domain that is thought to be responsible for the interaction and activation of CHMP7. At the sites where the membrane is intersected by microtubule bundles, LEM2 accumulates and undergoes liquid-phase separation, thereby triggering CHMP7 activation and ESCRT-III assembly [78]. Interestingly, the LEM2-CHMP7 system has been proposed to play a role as a sensor of local perturbations in the nuclear envelope barrier (see Section 4.2). ESCRT-III activity and recruitment at the NE are probably regulated by additional factors, like the CHMP4-binding protein CC2D1B, which prevents premature ESCRT-III and Spastin recruitment at the reforming NE. CC2D1B is thus believed to ensure timely polymerization of ESCRT-III at this organelle, necessary for proper NE regeneration [74].

ESCRT’s role in the sealing of the post-mitotic NE is evolutionarily conserved and has also been reported in lower eukaryotes. A similar mechanism mediating the re-establishment of nucleocytoplasmic compartmentalization during mitotic exit has been described during semi-open mitosis in *Schizosaccharomyces japonicus*, where orthologues of LEM2, CHMP7, CHMP4B, and VPS4 seem to play similar roles as those described in higher eukaryotes [79].

### 3.2. Sealing of the Nascent Autophagosome

Autophagy is a critical cellular process by which cytosolic components, from macromolecules to cellular organelles, are degraded in a controlled manner inside lysosomes [80]. This is essential for maintaining cell homeostasis and ensuring cell survival, allowing the removal of potentially harmful protein aggregates or damaged organelles. The best studied form of autophagy is macroautophagy (referred to as “autophagy” from now on), which delivers cytoplasmic material to lysosomes via a double-membrane organelle called the autophagosome [81]. The process starts with a double-membrane structure termed the phagophore that encloses bulk cytoplasm or specific cargo. The phagophore membrane, which is thought to come from different sources like the endoplasmic reticulum or the plasma membrane, eventually closes to form a complete autophagosome, resulting in engulfment of the cargo [80]. Then, the autophagosome fuses with a lysosome and the sequestered cargo is degraded by lysosomal hydrolases.

When the phagophore membrane has grown around cytoplasmic content and shaped, it needs to be closed to form a complete autophagosome [82]. Recently, several elegant imaging studies using advanced fluorescent probes have established a direct role for the ESCRT machinery in phagophore closure, during both starvation-induced autophagy and mitophagy [8,83,84,85]. Targeting of ESCRT-I components (VPS37A and VPS28) to the phagophore promotes the transient recruitment of ESCRT-III components, including CHMP2A and CHMP4B, bringing the two membranes of the phagophore leading edge in close proximity to allow membrane abscission. This is followed by VPS4-mediated depolymerization of ESCRT-III [83]. How ESCRT-I is recruited to the phagophore still remains to be fully understood, but studies in budding yeast mutants have suggested the involvement of the small endosomal GTPase Rab5 and Atg17, a subunit of the Atg1 autophagic kinase complex, as upstream regulators [85,86,87]. The essential role of ESCRTs in autophagosome sealing is evidenced by the fact that ESCRT depletion causes accumulation of autophagosomes that are incapable of fusing with lysosomes [88,89,90]. This is probably due to the failure of unsealed autophagosomes to recruit Syntaxin 17, a SNARE protein required for autophagosome-lysosome fusion [91].

## 4. ESCRTs in Membrane Repair

Most cellular membranes are exposed to damage, and different repair mechanisms are in place to promote cell survival by closing membrane holes or ruptures. Thanks to recent key work, the ESCRT machinery is now understood to repair damage in the plasma membrane, nuclear envelope, and throughout the endolysosomal network, carrying out important functions in membrane integrity maintenance.

### 4.1. Repair of the Damaged Plasma Membrane

Plasma membrane lesions can occur frequently as a consequence of numerous phenomena, including shear mechanical stress, pathogen assault, and chemicals. An efficient and rapid repair of plasma membrane damage is essential for cell survival. Repair is thought to be mediated by a variety of mechanisms, including patching by intracellular membranes or removal of the damaged area by outward budding and endocytosis [92]. These mechanisms are activated by common early signaling events, like influx of Ca^2+^ through the damaged plasma membrane, whereas downstream repair events seem to be dependent on the characteristics of the wounds.

Exocytosis of membrane-proxymal lysosomes and subsequent removal of wounded membrane is known to be a major repair mechanism for large lesions (200–500 nm) [93]. Membrane remodeling by the ESCRT complex has been shown to participate in the repair of small (<100 nm) but not large plasma membrane wounds [9,94]. Plasma membrane damage induced by mechanical force, detergents, pore-forming toxins, or laser wounding causes a rapid recruitment of ESCRT-III proteins to the site of damage, where they accumulate until wound closure [9]. This recruitment is followed by ESCRT-positive membrane budding and shedding, suggesting that ESCRTs may play a role in the detection and removal of small plasma membrane domains containing the site of damage. Damage-induced ESCRT recruitment is dependent on calcium and requires PDCD6, ALIX, and Annexin A7, indicating that these proteins could function as Ca^2+^ sensors that trigger recruitment [94,95] (Figure 3).

ESCRT-dependent membrane repair has been implicated in the resealing of endogenous pore-mediated plasma membrane damage during necroptosis [96], pyroptosis [97], and ferroptosis [98,99]. Moreover, recent work from the Mellman lab [100] has shown that the ESCRT machinery is involved in the repair of pores formed by perforin, a toxin released by cytotoxic T lymphocytes (CTLs) and natural killer cells to kill virus-infected and tumor cells. Ritter et al. visualized how ESCRT is recruited to sites of CTL engagement in cancer-derived cells immediately after perforin release. They also observed membrane protrusions containing ESCRT proteins within the cytolytic synapse, consistent with the previously proposed mechanism of membrane repair by vesicle shedding [9,94]. Inhibition of ESCRT machinery in cancer cells enhanced their susceptibility to CTL-mediated killing. Thus, repair of perforin pores by ESCRTs limits CTL-secreted granzyme entry into the cytosol, and potentially enables cancer cells to resist cytolitic T cell attack [100].

ESCRT-mediated PM repair has also been reported during interaction with fungal cells [101]. In response to candidalysin, a pore-forming peptide toxin secreted by *Candida albicans*, epithelial cells activate Ca^2+^-dependent repair mechanisms and dispose of damaged membrane regions by way of an Alg-2/Alix/ESCRT-III-dependent blebbing process.

It remains to be established if ESCRTs act in concert with other plasma repair mechanisms, and to define the spatiotemporal relationship of membrane wounding with Ca^2+^-regulated lysosomal exocytosis, up-regulation of endocytosis, and ESCRT recruitment. This will help to clarify how each of these pathways contributes to lesion removal and plasma membrane resealing, potentially revealing steps susceptible to therapeutic intervention [102].

### 4.2. Repair of Nuclear Envelope Ruptures

The location of the plasma membrane clearly makes it vulnerable to disruption, but internal membranes are also prone to damage. This is the case for the nuclear membrane, which due to the rigidity and large size of the nucleus, is particularly sensitive to damage when the cell moves through a confined space. In addition, reversible NE ruptures are frequently detected in laminopathies [103,104] or in cancer cells [77,105]. Loss of NE integrity is associated with the uncontrolled exchange of nucleo-cytoplasmic content, herniation of chromatin across the NE, and DNA damage, and may compromise cellular function and viability [77,103,104,105,106].

It has been shown that ESCRT helps to seal NE ruptures caused by mechanical forces imposed as cells migrate through constrictions [77,107] or other mechanical perturbations [108,109]. ESCRTs counteract Nesprin-2G-mediated cytoskeletal mechanical forces facilitating NE repair [110] and they also contribute to protecting the integrity of the micronuclei NE [111]. Whether the mechanisms mediating this ESCRT protective role are similar to those mediating ESCRT function at the NE in a physiological context still remains to be fully understood. However, important advances have been made in understanding how disruptions in the NE barrier are sensed, and new key players have been identified that help to seal the barrier [112].

Cells have the ability to monitor the integrity of the NE barrier and proper assembly of the nuclear pore complexes (NPCs). This surveillance mechanism is mainly formed by the ESCRT protein Chm7/CHMP7 and its inner nuclear membrane binding partner, Heh1/LEM2 [112]. At steady-state, these proteins are physically separated on either side of the NE, with CHMP7 localizing to the ER [113,114]. Any disruption in the nuclear-cytoplasmic organization will induce the physical association of LEM2 and CHMP7, which is thought to activate a repair mechanism to seal the NE. The details of how this happens still need to be fully defined. As mentioned above, in vitro studies show that the LEM2 winged helix domain directly binds to CHMP7 [78], inducing conformational changes on it and the formation of a CHMP7-LEM2 copolymer [78]. As in other ESCRT-mediated membrane fusion processes, probably additional ESCRTs such as CHMP4 and CHMP2A, alongside the ATPase Vps4, are also recruited to the NE (Figure 3). In support of this, Chm7 is known to directly bind Snf7/CHMP4 [5] and to be required for Snf7 and additional downstream ESCRTs (including Vps4) to be recruited to the nuclear envelope [7,39,75,113].

Recent work shows that over-stimulation of the CHMP7/LEM2 surveillance system may be deleterious to cell viability and directly contribute to DNA damage [111,113,115]. Thus, the CHMP7-LEM2 interaction must be tightly regulated, and mechanisms might exist to prevent these two proteins from aberrantly interacting. CC2D1B is one of the proposed regulators of CHMP7 function at the NE at the end of mitosis [74] (Figure 3). More recently, the mitotic kinase CDK1 has been shown to phosphorylate CHMP7 upon mitotic entry, rendering CHMP7 unable to interact with LEM2 [114]. This suggests a possible mechanism that prevents CHMP7-LEM2 association when the nuclear envelope is disassembled in prophase. Local CHMP7 dephosphorylation at the nascent nuclear envelope might license the LEM2-CHMP7 interaction that triggers ESCRT-III recruitment to reseal the NE [114]. The Chm7-LEM2 interaction can also be regulated by Chm7´s binding to phosphatidic acid-rich membranes [116] and Hub1-mediated alternative splicing of LEM2 [117].

The CHMP7/LEM2 system most probably mediates the sealing of small NE holes (<100 nm). Several studies support this hypothesis, such as the observation of membrane necks of ~50 nm diameter upon Chm7 hyperactivation [113], the formation of in vitro CHMP7-LEM2 polymers of around that diameter [115], and the fact that ESCRTs are found in ~30–50 nm holes at the reforming NE [6]. Regarding the repair of larger ruptures, recent work has uncovered a key role for barrier-to-autointegration factor (BAF) in repairing mechanically induced ruptures in mammalian cells [109]. Upon exposure of genomic DNA to the cytosol, a non-phosphorylated cytoplasmic pool of BAF is recruited and binds nuclear DNA to localize to sites of nuclear rupture rapidly and transiently. BAF is required to repair the NE via subsequent recruitment of LEM-domain proteins at rupture sites (Figure 3). CHMP7 is also recruited, but it is in fact dispensable for the efficient repair of these mechanically induced ruptures [109]. Although the ESCRT-III complex may facilitate the repair process, likely by sealing small holes in terminal steps, there must be other yet-to-be-characterized mechanisms necessary to repair this kind of NE rupture.

### 4.3. Repair of Damage in the Endolysosomal Membrane

Cells take in extracellular material through the endolysosomal network to generate nutrients, clear debris, and sample their environment. Wounds in the membranes of the endosomes, phagosomes, and lysosomes that comprise this network are caused by pathogens, particulates, and other chemical or metabolic stresses [118]. Consequences for cell health vary depending on the location and extent of the damage. Lysosomes are one the organelles more exposed to membrane damage and extensive lysosomal membrane permeabilization is known to trigger death after release of enzymes into the cytosol [119,120].

Severely damaged lysosomes are removed by a selective autophagy process termed lysophagy [121,122,123]. This process, although crucial for maintaining cellular homeostasis, is inherently slow and accompanied by leakage of cytotoxic material into the cytoplasm [118]. Several studies over the past five years support the hypothesis that limited damaged to the lysosome membrane can be rapidly repaired by the ESCRT machinery to restore compartmental integrity [10,11,124,125]. L-leucyl-L-leucine methyl ester (LLOME) and glycyl-L-phenylalanine 2-naphthylamide (GPN)—the compounds used to study lysophagy—damage lysosomes within a minute after addition to cells. This triggers rapid recruitment of ESCRT-I, ALIX, and most ESCRT-III proteins together with various partners, including VPS4, to damaged organelles [10,11,126]. Damage by LLOME or GPN is reversible within minutes after washout of the drug [10,127]. Depletion of TSG101 and ALIX slows or completely blocks this rapid recovery, implicating ESCRT function in lysosomal repair [10]. However, recent studies showed a sustained presence of ALIX and associated ESCRT proteins on organelles damaged by prolonged exposure to LLOME [125]. This suggests that ESCRTs may contribute not only to immediate but also to delayed responses to lysosomal damage. Consistent with a role for the ESCRT pathway in endolysosomal membrane repair, knocking down of ESCRT components leads to membrane damage and leakiness of the endolysosomal compartment and has been reported to enhance prion-like propagation of tau aggregation, a critical step in the progression of neurodegenerative diseases [124].

Similarly to damage to the plasma membrane, Ca^2+^ efflux from damaged lysosomes and the Ca^2+^ sensor PDCD6 have been reported to trigger ESCRT recruitment to ruptured lysosomes [10,125]. However, the mechanism of ESCRT-III recruitment seems to be more complex than that to the damaged plasma membrane, and involves not only the Ca^2+^ and sensor PDCD6-binding protein ALIX [10,11], but also additional factors [128] (Figure 3). One of them is GAL3, which interacts with ALIX and is required for efficient recruitment of ALIX and ESCRT-III to damaged lysosomes. It is thought that GAL3 could provide a later and more sustained signal than Ca^2+^ efflux, and it has the additional function of promoting lysophagy in the case of more severe lysosome damage [125]. An additional factor could be the Parkinson´s-disease-related leucine-rich repeat kinase 2 (LRRK2), which upon lysosome damage triggers the recruitment of the small GTPase Rab8A and subsequently the ESCRT-III protein CHMP4B to damaged organelles in macrophages [129]. Interestingly, Ca^2+^ efflux can also trigger additional ESCRT-independent lysosomal repair mechanisms mediated by annexins A1 and A2 [130] and sphingomyelin scrambling and turnover [131].

Once at the site of damage, it is still unknown how ESCRTs function to repair wounds in the lysosome membrane. Given the known topology of other ESCRT-regulated processes, it is possible that the membrane-remodeling performed by ESCRT-III filament spirals acts to shed damaged membranes into the lumen of the lysosome for recycling (Figure 3). Alternatively, it is believed that injuries in membranes can spontaneously reseal if the wound is small enough [132,133], so ESCRT-III spirals could also act to prevent nanoscale damage from expanding, allowing the membrane to spontaneously reseal. Further research in both model systems and live cells will be crucial to learn more about the specific damage signals that trigger ESCRT recruitment to damaged endolysosomal membranes and fully elucidate the mechanistic role of ESCRT in membrane repair.

Recent work suggests that ESCRT can play a role not only in restoring lysosomal integrity after damage, but also in the physiological regulation of lysosomal morphology, mediating the degradation of lysosomal membrane proteins [134,135].

## 5. Perspectives

Since their discovery, investigation into ESCRT proteins has been a very active and continuously expanding field of research. The ESCRT machinery is now appreciated as a highly versatile membrane remodeling complex that is used by the cell in a number of essential processes to effect a topologically unique membrane scission. However, for many ESCRT-dependent processes we still do not have a complete picture of how ESCRTs are recruited or how they cut membranes. In recent years, extensive efforts have been made to elucidate the structure and behavior of ESCRT filament assemblies and understand exactly how ESCRT proteins catalyze membrane separation. Although important questions remain open, key advances have been made and the field seems finally to be converging into a unifying molecular mechanism to explain this membrane remodeling activity.

In recent years, an additional role of ESCRTs in controlling membrane integrity has emerged. The molecular mechanisms mediating ESCRT sealing and repair of damaged membranes need to be fully defined. A better understanding of this process, which is key for cell viability and even exploited by certain pathogens, will be relevant to potential manipulations of membrane sealing for therapeutic applications.

## Figures and Tables

**Figure 1 membranes-12-00633-f001:**
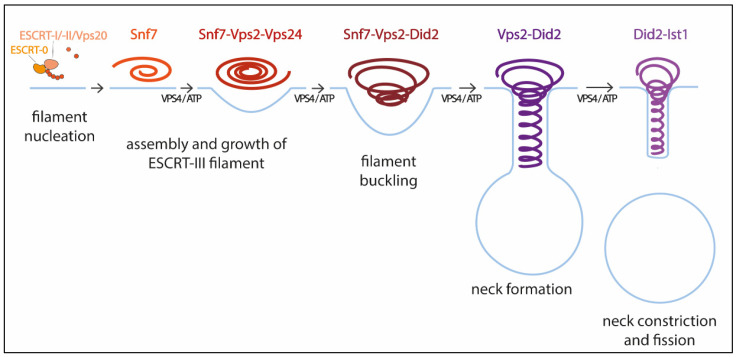
Model for membrane constriction and fission driven by ESCRT-III filament assembly and disassembly. The figure illustrates the sequential recruitment of ESCRT-III components, polymerization, and replacement of different filament subunits driven by Vps4, resulting in constriction and final scission of the membrane (adapted from [59]).

**Figure 2 membranes-12-00633-f002:**
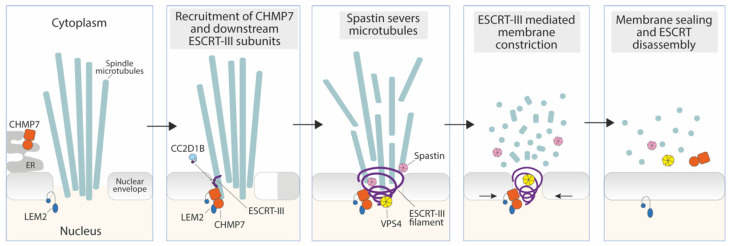
ESCRT-mediated sealing of the nuclear envelope during mitotic exit. In late anaphase, the inner nuclear membrane protein LEM2 recruits and activates CHMP7 to holes in the reforming nuclear envelope, which drives ESCRT-III polymerization. Premature ESCRT-III recruitment to the nuclear membrane is prevented by CC2D1B, which dissociates prior to ESCRT recruitment. Spastin recruitment by ESCRT-III triggers severing of mitotic spindle microtubules, while VPS4 remodeling of ESCRT-III filaments promotes membrane constriction and sealing of the nuclear envelope.

**Figure 3 membranes-12-00633-f003:**
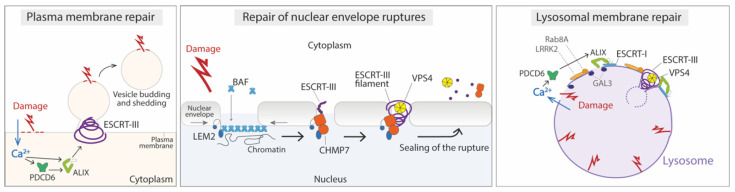
Membrane repair processes mediated by ESCRT. Left panel: plasma membrane repair. Entry of calcium through the damaged membrane triggers rapid ESCRT recruitment, mediated by PDCD6 and ALIX. ESCRTs are thought to promote membrane budding and shedding of small domains containing the site of damage; middle panel: nuclear envelope (NE) repair. Upon NE rupture, cytosolic BAF coats the exposed chromatin and interacts with LEM2, facilitating the recruitment of nuclear membrane and the interaction with CHMP7. CHMP7 subsequently promotes the nucleation and polymerization of ESCRT-III, which together with VPS4 constricts the rupture and promotes sealing; right panel: lysosome repair. After damage, calcium efflux from the lysosome promotes ESCRT-I and -III recruitment through PDCD6, ALIX, and probably other factors like GAL3 and LRRK2, which phosphorylate the small GTPase Rab8A. It is thought that the membrane-remodeling performed by ESCRT-III filament spirals acts to shed damaged membranes into the lumen of the lysosome for recycling.

**Table 1 membranes-12-00633-t001:** ESCRT complexes and their protein subunits in yeast and humans.

Complex	*S. cerevisiae*	*H. sapiens*
**ESCRT-0**	Vps27	HGS (HRS)
	Hse1	STAM1, STAM2
**ESCRT-I**	Vps23 (Stp22)	TSG101
	Vps28	VPS28
	Vps37 (Srn2)	VPS37A/B/C/D
	Mvb12	MVB12A/B, UBAP1, UBA1L, UMAD1
**ESCRT-II**	Vps22 (Snf8)	EAP30 (SNF8)
	Vps25	EAP20 (VPS25)
	Vps36	EAP45 (VPS36)
**ESCRT-III**	Did2 (Vps46, Chm1)	CHMP1A/B
	Did4 (Vps2, Chm2)	CHMP2A/B
	Vps24 (did3)	CHMP3
	Snf7 (Vps32, Did1)	CHMP4A/B/C
	Vps60 (Chm5)	CHMP5
	Vps20 (Chm6)	CHMP6
	Chm7	CHMP7
	Ist1	IST1
**ESCRT-associated**	Vps4	VPS4A/B (SKD1)
	Vta1	VTA1 (LIP5, DRG-1)
	Bro1 (Vps31)	ALIX (PDCD6IP), HD-PTP (PTPN23)
	Doa4	UBPY, STAMBP

Alternative protein symbols are shown in parentheses.

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
