# Peer review of "The ESCRT Machinery: Remodeling, Repairing, and Sealing Membranes"

_membranes, 2022, doi:10.3390/membranes12060633_

Round 1

Reviewer 1 Report

This review is of very high quality and interesting content, written by one of the experts in the field of ESCRT. It is acceptable, but I just suggest including the following paper about the role of Annexin A1 and A2 in ESCRT-independent repair of endolysosomal membranes.

Annexins A1 and A2 are recruited to larger lysosomal injuries independently of ESCRTs to promote repair. (FEBS Lett. 2022 Apr;596(8):991-1003. doi: 10.1002/1873-3468.14329. )

Author Response

I thank the reviewer for the positive assessment of the manuscript. I have included now the reference Yim et al, FEBS Lett. 2022 [Ref. No. 131] and mentioned it in the text [lines 444 to 446]. I have also included a reference to another ESCRT-independent lysosomal repair mechanism mediated by sphingomyelin scrambling and turnover (Niekamp et al, Nat Commun. 2022) [Ref. No. 132].

Reviewer 2 Report

This manuscript discusses the recent advances in the molecular mechanisms that underlie membrane constriction and scission by ESCRT-III and describes this machinery's involvement in the sealing and repairing of damaged cellular membranes. The manuscript is well written and discussed. However, the author needs to add more recent references and related content in the revised manuscript. For example, there are 55 references before 2017 (55/114), while only 59 references reported in the last five years were reviewed in this manuscript.

Author Response

I thank the reviewer for the positive assessment of the manuscript. 23 new references from the last five years have now been added to the manuscript (highlighted in red in the References section). Most of them were published in 2021 and 2022 so I believe this has improved the original manuscript and helped to fully cover now the most recent advances in the field.